# Effects of Resveratrol on In Vivo Ovarian Cancer Cells Implanted on the Chorioallantoic Membrane (CAM) of a Chicken Embryo Model

**DOI:** 10.3390/ijms25084374

**Published:** 2024-04-16

**Authors:** Kenny Chitcholtan, Melanie Singh, Alex Tino, Ashley Garrill, Peter Sykes

**Affiliations:** 1Gynaecological Cancer Research Group, Department of Obstetrics and Gynaecology, University of Otago Christchurch, Christchurch 8011, New Zealand; alex.tino@postgrad.otago.ac.nz (A.T.); peter.sykes@otago.ac.nz (P.S.); 2School of Biological Sciences, University of Canterbury, Christchurch 8140, New Zealand; melanie0singh@gmail.com (M.S.); ashley.garrill@canterbury.ac.nz (A.G.)

**Keywords:** ovarian cancer, resveratrol, chicken embryo, NF-κB, SLUG, cell invasion, collagen scaffold, SKOV-3, OVCAR-8, VEGF

## Abstract

Ovarian cancer poses a significant threat to patients in its advanced stages, often with limited treatment options available. In such cases, palliative management becomes the primary approach to maintaining a reasonable quality of life. Therefore, the administration of any medication that can benefit patients without a curative option holds potential. Resveratrol, a natural compound known for its in vitro anticancer activities, has generated contrasting results in vivo and human studies. In this study, we aimed to assess the anticancer effects of resveratrol on ovarian cancer cells grown on the chorioallantoic membrane (CAM) of chicken embryos. Two ovarian cancer cell lines, OVCAR-8 and SKOV-3, were cultured in collagen scaffolds for four days before being implanted on the CAM of chicken embryos on day 7. Different doses of resveratrol were applied to the CAM every two days for six days. Subsequently, CAM tissues were excised, fixed, and subjected to histological analysis. Some CAM tumours were extracted to analyse proteins through Western blotting. Our findings indicate that specific doses of resveratrol significantly reduce angiogenic activities, pNF-κB levels, and SLUG protein levels by using immunohistochemistry. These results suggest that resveratrol may have the potential to impact the behaviour of ovarian cancer CAM tumours, thereby warranting further consideration as a complementary treatment option for women with incurable ovarian cancer.

## 1. Introduction

Ovarian cancer often manifests at an advanced stage, characterised by the presence of tumour nodules of varying sizes on the surface of the peritoneal cavity, including the abdominal wall, omentum, mesentery, and intestinal tracts. Unfortunately, patients diagnosed at this disease stage face a grim prognosis, as the tumours exhibit resistance to currently available chemotherapeutic regimens. Consequently, it becomes imperative to identify a non-toxic alternative compound that shows promise as a potential treatment option during palliative management. The primary objectives of such treatment would be to enhance the patient’s quality of life and significantly prolong their survival.

Resveratrol, a plant-based compound found in various plants, has garnered significant attention in cancer research due to its diverse antitumour activities that have been observed under in vitro conditions. One notable aspect of its antitumour activity is its ability to modulate the NF-κB signalling pathway, which impacts tumour cell viability, angiogenesis, growth, and metastasis [1]. The wide range of antitumour effects associated with resveratrol’s influence on cell signalling underscores its potential as a promising compound for cancer treatment. Resveratrol has shown antitumour activities in the cell monolayers of ovarian cancer [2] and the inhibition of ovarian cancer cell invasion mediated by LPA-EGFR pathways [3]. Furthermore, the combination of lysophosphatidic acid (LPA) and hypoxia-mediated vascular endothelial growth factor (VEGF) was abolished by resveratrol by interrupting the p42/p44 MAPK and p70S6K signalling in cell monolayers [4]. Resveratrol reduces cell proliferation by interrupting the PI-3K/Akt and ERK signalling pathways in OVCAR-3 and ascitic fluid-derived primary ovarian cancer cells [5]. A combination of resveratrol and platinum agents significantly increased antitumour activities compared to a single treatment alone [6]. The dose dependence of resveratrol showed tumour growth inhibition in the ovarian tumour xenograft model via intraperitoneal injection of the resveratrol [7].

The biological effects of resveratrol are observed in a tissue- and dose-dependent manner, making it challenging to draw definitive conclusions regarding its impact on cellular behaviour. Previous studies have presented conflicting results regarding the antitumour activities of resveratrol in both in vitro and in vivo cancer models [8]. For instance, while resveratrol reduced in vitro cell invasion in the NuTu-19 rat ovarian cancer cell line, it exhibited no discernible effect in vivo [9]. Resveratrol is cytotoxic in a time- and dose-dependent manner in prostate cancer cell lines in vitro, but there is no difference in growth among the control and resveratrol-treated tumour mouse xenografts [10]. Moreover, resveratrol exhibits inhibition of the LNCaP prostate cancer cell line by modulating both androgen- and estrogen-mediated pathways, demonstrated both in vitro and in vivo. However, exposure to resveratrol also results in increased angiogenesis and the inhibition of apoptosis in vivo studies [11].

The development of a preclinical model capable of replicating the complex microenvironment of advanced ovarian cancer in humans is essential for elucidating the mechanisms underlying the molecular and cellular diversity of ovarian cancer cells. Advanced ovarian cancer often manifests as multiple tumour nodules residing on the surfaces of the peritoneal cavity and internal organs. These tumour nodules exhibit well-developed vasculature networks, facilitating the delivery of nutrients and oxygen crucial for sustaining cell growth and survival. Given the significance of the unique peritoneal microenvironment in advanced ovarian cancer, the utilisation of a similar preclinical in vivo model is pivotal for advancing ovarian cancer research. The chicken embryo chorioallantoic membrane (CAM) represents an extracorporeal membrane that serves as a gas exchange surface during embryo development, supported by a dense capillary network [12]. By day 8 to 10, the CAM reaches full development, becoming capable of sustaining tissue grafts [13,14]. Histologically, the CAM comprises three germ layers: ectoderm, mesoderm, and endoderm. The ectoderm layer is composed of epithelial cell layers; the mesoderm layer consists of stromal compartments, including connective tissue, blood vessels, and collagens, while the endoderm layer comprises a single layer of epithelial cells [15]. Although the utilisation of CAM as a preclinical tool in scientific research has been recognised for decades, its application in oncology research has lagged behind the use of rodent models [16]. Despite its potential, the popularity of the CAM in ovarian cancer research remains limited, despite its occasional use in select studies [17,18]. 

It appears the CAM has enough similarity to the ovarian cancer microenvironment to offer a promising avenue to assess the potential effectiveness of resveratrol within the context of ovarian cancer. In this study, our hypothesis posits that resveratrol could potentially influence the growth of ovarian tumours in the CAM model. We further propose that the modulation of growth may be linked to specific doses of resveratrol, thereby impacting the signalling proteins within the tumour cells.

## 2. Results

### 2.1. Morphology of Tumour Implants Grown on CAMs

Both OVCAR-8 and SKOV-3 implants were subjected to different doses of resveratrol treatment, and subsequent imaging was performed to evaluate the vascular networks. Figure 1 illustrates tumour implants derived from OVCAR-8 cells, displaying vascular networks in both the control and resveratrol-treated tumours. 

Additionally, beneath the tumour implants of resveratrol-treated OVCAR-8 cells, distinct vascular networks were clearly observed. The tumour implants are associated with several large blood vessels, accompanied by a network of small vascular branches (Figure 2).

In SKOV-3 tumour implants, noticeable vascular networks were present surrounding the tumour implants (Figure 3). Furthermore, vascular networks were also observed underneath the tumour implants (Figure 4). 

### 2.2. Resveratrol Affects Vascularisation in CAM Tumour Implants

The number of red blood cells detected within the tumour implants could serve as an indicator of the extent of angiogenesis, as a more pronounced angiogenesis process typically results in more red blood cells. Moreover, the application of haematoxylin and eosin staining enables red blood cells to be visually distinguished from cancer cells. In Figure 5 (left), cross-sectional views of OVCAR-8 tumour implants treated with varying doses of resveratrol are displayed. Significantly fewer red blood cells per square millimetre were observed in the implants treated with 45.6 µg (1 mM) of resveratrol in comparison to the control implant (*p* < 0.0001) (Figure 5, right).

Similarly, the SKOV-3 tumour implants exhibited a significant decrease in the number of red blood cells when treated with 91.24 µg (2 mM) of resveratrol (*p* = 0.012) (Figure 6 right).

### 2.3. Resveratrol Did Not Affect Immunohistochemistry Levels of Vascular Endothelial Growth Factor (VEGF) or Ki67 in Tumour Implants

To assess the impact of resveratrol on VEGF levels and subsequent angiogenesis in OVCAR-8 and SKOV-3 tumour implants, we performed immunohistological staining using an anti-VEGF antibody. The staining revealed discrete red-brown signals indicative of VEGF presence within the tumour implants of both OVCAR-8 (Figure 7, top-left) and SKOV-3 (Figure 7, bottom-left). However, no statistically significant reduction in VEGF levels was observed with the concentrations of resveratrol that resulted in a decrease in the number of red blood cells within the tumour implants (Figure 7, top-right and bottom). These findings may suggest that the inhibition of reduced red blood cells, serving as an indirect indication of angiogenic establishment in the CAM, is not associated with the level of tumour VEGF.

In addition, we conducted immunohistochemical staining of Ki-67 proteins in the tumour tissue implants treated with selective doses of resveratrol. Our findings revealed that Ki-67 expression remained unchanged in resveratrol-treated tumours compared to the control group (Appendix A). These results suggest that resveratrol may not affect the growth modulation of tumour cells.

### 2.4. Resveratrol at Selective Dose Affected pNF-κB Activation and Cancer Cell Invasion in the Histological Sections of Tumour Implants

In our investigation, we examined the impact of resveratrol on NF-κB expression and its phosphorylation within tumour implants. Previous studies have demonstrated that resveratrol can influence NF-κB and its phosphorylation in ovarian cancer cells under in vitro conditions [19]. Figure 8 presents the levels of NF-κB in OVCAR-8 (Figure 8A) and SKOV-3 (Figure 8C) tumour implants, indicating no significant modulation by resveratrol compared to the control. However, at a concentration of 91.24 µg (2 mM), resveratrol significantly reduced the levels of phosphorylated NF-κB (pNF-κB) in both the OVCAR-8 (Figure 8B, *p* = 0.0004) and SKOV-3 (Figure 8D, *p* = 0.018) tumour implants. Interestingly, in SKOV-3 tumour implants, the dose of 45.65 µg (1 mM) of resveratrol significantly increased the levels of pNF-κB (Figure 8D, *p* = 0.007). 

Additionally, we conducted an evaluation of the impact of resveratrol on the invasion of tumour cells within the CAM tissue. The invasive cancer cells were quantified by analysing high-resolution images obtained from a light microscope. Figure 9 demonstrates that resveratrol at a concentration of 91.24 µg (2 mM) significantly decreased the number of invasive cancer cells in both OVCAR-8 (Figure 9A, *p* = 0.0056) and SKOV-3 (Figure 9B, *p* = 0.001) within the CAM tissue.

### 2.5. Resveratrol Does Not Change Total Protein Level of PCNA, NF-κB, pNF-κB, and SLUG

Subsequently, we proceeded to extract proteins from the OVCAR-8 tumour CAM tissue and subjected them to analysis using Western blotting techniques (Figure 10). Our findings revealed that the proliferative protein marker PCNA (Figure 10A), NF-κB (Figure 10B), phosphorylated NF-κB (pNF-κB) (Figure 10C,D), and SLUG (Figure 10E) did not exhibit a decrease in expression when treated with 91.24 µg (2 mM) of resveratrol. Similarly, analogous results were obtained from the SKOV-3 tumour CAM tissue (Figure 11A–F).

### 2.6. Immunohistological Fluorescent Staining of SLUG Protein: Resveratrol Significantly Reduced SLUG Proteins in Cancer Cells in the CAM Tissue

Figure 9A,B demonstrate a decrease in the number of invasive cancer cells within the CAM tissue, suggesting reduced invasiveness. Considering the association of the SLUG protein with increased cell invasiveness, our investigation aimed to evaluate SLUG protein expression through tissue immunofluorescent staining of tissue sections. Our objective was to assess whether different doses of resveratrol could potentially reduce the protein level at a single-cell level, thereby attenuating cell invasion. Figure 10 and Figure 11 illustrate that the total protein levels detected by Western blot analysis of SLUG were not reduced, which may be attributed to the presence of SLUG in the CAM itself. Hence, the analysis of histological fluorescent staining of SLUG at a single-cell level is highly sensitive, and our focus was to measure the antigen intensity specifically in cancer cells, which can be distinguished from SLUG-associated CAM cells. 

The results revealed a significant reduction in the SLUG protein at the single-cell level (Figure 12A, *p* = 0.0176) in OVCAR-8 tumours treated with 91.24 µg (2 mM) of resveratrol. Furthermore, in SKOV-3 CAM tumours, all experimental doses of resveratrol significantly reduced the intensity of SLUG protein staining (Figure 12B, *p* = 0.0001 (45.6 µg (1 mM)), *p* = 0.0001 (91.24 µg (2 mM)), *p* = 0.0022 (182.4 µg (4 mM))). These findings suggest that resveratrol treatment at specific doses effectively decreased the expression of SLUG protein, potentially contributing to the reduction in cell invasion observed within the CAM tissue.

## 3. Discussion

This study investigated the effect of resveratrol on the tumour behaviour of ovarian cancer implanted in the CAM model. Complex vascular networks surround the CAM ovarian tumour implants of OVCAR-8 and SKOV-3 lines. These vascular networks in the CAM model are akin to tumour nodules found on the surface of the human peritoneal cavity, suggesting the physiological environment could have a considerable impact on the behaviour of ovarian tumours [20]. Collectively, the microscopic images of tumour implants in the CAM model are similar to tumours found in ovarian cancer patients, where ovarian tumour nodules are surrounded by visible vascular networks on peritoneal surfaces.

In our study, the OVCAR-8 and SKOV-3 CAM tumours exhibit a dense network of blood vessels projecting from the CAM, evident in both control and resveratrol-treated tumours. No significant differences were observed between the controls and treatments under stereomicroscopic examination. Histological analysis, focusing on the presence of red blood cells as an indirect indicator of vascular networks in the CAM tumours affected by resveratrol treatment, revealed an intriguing finding. Specifically, a distinct resveratrol dosage led to a reduction in the number of red blood cells in OVCAR-8 (45.6 µg (1 mM)) and SKOV-3 (91.24 µg (2 mM)). At this juncture, we lack a plausible explanation for why these particular doses, as opposed to others, elicit a biological effect. One plausible hypothesis is that these doses, by exerting a biological impact, may attain a concentration threshold optimised for the biological functions of proteins responsible for cellular function. The development of blood vessels around the CAM tumour is triggered by the secretion of VEGF proteins, primarily originating from the tumour itself. Notably, the particular doses of resveratrol that markedly reduced red blood cells in both OVCAR-8 and SKOV-3 (Figure 7) did not concurrently decrease the levels of VEGF, as determined through immunohistological staining in the CAM tumour. These findings may indicate that VEGF is not directly linked to the recruitment of blood vessels in the CAM tumour. One plausible explanation for the lack of a VEGF reduction in resveratrol-treated tumours could be the direct inhibition of endothelial cells within the vascular capillaries on the CAM by resveratrol. This concept has been explored in several previous studies. For instance, resveratrol has been shown to inhibit the proliferation and migration of vascular endothelial cells through the activation of eukaryotic elongation factor-2 kinases both in vitro and in vivo [21]. Additionally, the compound hinders endothelial cell tube formation and migration by suppressing glycolytic proteins [22]. An intriguing study, conducted by Dias et al. (2008), revealed that resveratrol, at various concentrations, significantly inhibits early vessel formation, leading to a reduction in vitelline vessels in chicken embryos. Unexpectedly, it also resulted in an increase in the body length of the embryos [23].

We previously reported that in vitro studies, resveratrol reduced the growth of and the levels of the VEGF protein in SKOV-3 and OVCAR-8 cell spheroids in a concentration-dependent manner [24]. We also showed that a 6-day treatment regimen with resveratrol reduced VEGF but increased IL-8 in SKOV-8 ovarian cancer cell aggregates [19]. Other studies have shown that resveratrol decreases the gene and protein expression of VEGF and MMP-9 in endometrial stromal cells of patients with endometriosis [25]. Furthermore, resveratrol has been shown to significantly reduce the VEGF and TNF-α genes and proteins in a randomised exploratory clinical trial [26]. Resveratrol treatment in endometrial stromal cells from patients with endometriosis leads to a time-dependent reduction in IL-8 [27]. Chronic resveratrol treatment for five weeks in fibroblast cells also demonstrates a significant decrease in IL-8 levels [28]. Moreover, in adipocytes under inflammatory conditions, resveratrol reduces IL-8 levels in a concentration-dependent manner by inhibiting the NF-κB protein [29,30].

The impact of resveratrol on mammalian cells can be variable, as evident in both in vitro and in vivo investigations. The lack of consistency across studies underscores the need for cautious interpretation, considering tissue-specific and resveratrol-dose-specific events. Resveratrol is widely recognised for its hormetic properties in both in vitro and in vivo studies [31]. Hormesis refers to the dual-phase response of cells or organisms to a chemical, characterised by stimulation at low concentrations and inhibition at high concentrations. These distinctive properties of resveratrol are contingent upon the cell type and concentration. For example, Juhasz et al. conducted a study involving rats fed three different doses of resveratrol (2.5 mg/kg, 25 mg/kg, and 100 mg/kg) for up to 30 days, revealing a hormetic response where resveratrol exhibited cardioprotective effects at lower doses and detrimental effects at higher doses [32]. Additionally, research by Plauth et al. demonstrated that under physiologically relevant conditions, the primary biological effects of resveratrol could be ascribed to its induction of oxidation through the reprogramming of Nrf2-specific gene expression [33].

Due to the hormetic characteristics of resveratrol, preclinical and clinical studies have yielded conflicting results regarding its use in human subjects [34]. Compounding these challenges, resveratrol exhibits poor bioavailability and instability in humans [35]. Additionally, there are issues surrounding the detection methods of resveratrol in serum and tissues, which may not accurately reflect its biological properties [36]. Our study grapples with the question of how resveratrol bioavailability in CAM tissue might limit its biological function. Resveratrol, as used in our study, dissolves in a 5 mM glucose solution, posing solubility issues. We cannot use DMSO as a solvent for dissolving resveratrol in chicken embryos due to its toxicity. Zhong’s study suggests that intraperitoneal administration of resveratrol dissolved in DMSO exhibits higher drug bioavailability and more substantial tumour-suppressing effects than resveratrol dissolved in ethanol in rat orthotopic ovarian cancer models [37]. The CAM itself comprises selectively permeable cell layers. Drugs applied topically onto the CAM can be absorbed through the membrane, reach systemic circulation, and influence the body development of the chicken embryo [14]. This route of drug administration in the CAM is analogous to the intraperitoneal route in rodents and humans, where drugs are absorbed by mesothelial cell layers covering the peritoneal membrane in the abdominal cavity. In our study, specific doses of resveratrol may reduce efficacy due to solubility issues and consequently lower bioavailability. However, these factors alone cannot account for the differing efficacy between two cell lines at similar concentrations as shown in Figure 12. Therefore, we posit that the cell-type dependence, which encompasses molecular diversity among them, is more critical than concentration bioavailability alone in a CAM model.

Numerous studies have revealed the hormetic activities of resveratrol in diverse cell types. For example, resveratrol reduces apoptosis and activates the HIF-1α/VEGF pathway to promote angiogenesis after three days of implantation of duck ovarian tissues in the renal capsule tissue of the recipient duckling [38]. Different concentrations of resveratrol promote a change in VEGF expression in human umbilical vein endothelial cells (HUVECs) alone, and a co-culture with leukemic cancer cells (Jurkat cells) is dependent on the cell type and concentration, adding the complexity of the resveratrol mechanistic action upon the origin of tissue specificity [39]. The administration of resveratrol to cerebral endothelial cells stimulates tube formation in Matrigel assays, indicating a crucial role for resveratrol in inducing neurovascular recovery after a stroke [40]. Moreover, resveratrol demonstrated an ability to augment myocardial angiogenesis, both in vivo and in vitro, by inducing VEGF expression through the regulation of thioredoxin and heme oxygenase-1 proteins [41]. The cellular effects of resveratrol, as demonstrated in our study, appear to be dose-dependent and corroborated by numerous other studies. For example, a specific dose of resveratrol affects the mineralisation of human and rat adipose-derived stem cells (ADSCs) to promote osteogenic differentiation [42]. A low concentration of resveratrol profoundly inhibits the cell invasion of 4T1 tumour spheroids with a co-culture of embryonic stem cells via the reduction in the tumour’s MMP-9 expression [43]. The different concentrations of resveratrol have bifunctional effects in an estrogen-dependent manner on the ErbB2 levels in human breast cancer cells [44].

Our findings reveal a significant decrease in the immunohistological staining of pNF-κB in OVCAR-8 (Figure 8B) and SKOV-3 (Figure 8D) cells treated with 91.24 µg (2 mM) of resveratrol at the single-cell level. Furthermore, at this specific dose, there is a notable reduction in tumour cells observed in the CAM tissue (Figure 9), suggesting a potential association between NF-κB protein activation and cancer cell invasion/migration. Given the multifaceted nature of resveratrol’s effects on cellular targets, it is not surprising that its antitumour activities are dependent on tissue type and dosage. There are discrepancies between in vitro and in vivo studies regarding resveratrol and its ability to modulate cancer cell metastasis. For instance, in tissue-specific resveratrol activities, resveratrol supplementation has been shown to delay the development and decrease the metastasising capacity of spontaneous mammary tumours in HER-2/neu transgenic mice [45]. However, these effects do not appear to impact in vivo NuTu-19 ovarian cancers in rats [9]. High doses of 50 and 100 mg/kg resveratrol inhibit the growth of ovarian cancer PA-1 cell xenograft tumours associated with reduced eEF1A2 expression [7]. While resveratrol showed a significant association with worsened survival in LAPC-4 prostate tumours, its impact on survival remained unchanged in LNCaP prostate tumours [46]. Another study indicated that, in vitro, resveratrol extended growth inhibitory effects on cultured LNCaP cells through multiple pathways, including steroid hormone-dependent pathways. However, in vivo, resveratrol delayed the initial development of xenograft LNCaP cell tumours, yet exposure to resveratrol appeared to promote angiogenesis and inhibit apoptosis [11]. Furthermore, although in vitro data suggest promising antitumour activity of resveratrol in prostate cancer and melanomas, its efficacy in inhibiting tumour growth is limited [10]. Low concentrations of resveratrol have shown the potential to promote breast cancer growth and metastasis in immunocompromised mice. This underscores the importance of exercising caution and considering the implications of resveratrol’s use as an alternative compound for breast cancer patients [47].

One significant finding of our study is that resveratrol, at a dosage of 91.24 µg (2 mM), significantly decreased the immunofluorescent levels of SLUG proteins at the single-cell level in OVCAR-8 tumours (Figure 12A). Impressively, all the doses of resveratrol used also significantly reduced SLUG levels in SKOV-3 tumours (Figure 12B). This suggests that one potential cellular target of resveratrol may be to suppress cancer cell invasion and migration without necessarily affecting tumour growth. The previous study demonstrated that resveratrol exhibits a dose-dependent suppression of migration-promoting adhesion proteins, p65-NF-κB and SLUG, in colorectal cancer cell lines [48]. Additionally, high doses of resveratrol have been shown to reduce the expression of SLUG in vitro and to inhibit the migration of the human breast cancer MDA231 tumour xenograft model [49]. Resveratrol has also been found to inhibit ovarian cancer cell invasion, where lysophosphatidic acid (LPA) activates EGFR through Gi and G13, subsequently inducing Ras/Rho/ROCK signalling [3]. Moreover, resveratrol has been observed to inhibit LPA-induced cell migration in ovarian cancer cells by restoring autophagy, which in turn triggers cell death mediated in response to platinum [50].

One avenue that our study has not yet explored is the potential combination of resveratrol with other cytotoxic agents or with other food compounds. There are numerous reports highlighting these promising findings. For instance, co-treatment of resveratrol with low doses of cisplatin or carboplatin has been shown to inhibit the regrowth of ovarian cancer cells by inducing apoptosis and autophagy cell death [6]. Resveratrol has also been found to enhance the therapeutic efficacy of temozolomide (TMZ) by inhibiting TMZ-induced autophagy cell survival in brain tumours both in vitro and in vivo [51]. Additionally, the combination of resveratrol and doxorubicin has been demonstrated to inhibit tumour growth associated with NF-κB both in vitro and in vivo in breast cancer [52]. The combination of resveratrol with other food compounds and chemotherapeutic agents shows a strong suppression of VEGF-mediated angiogenesis in the colorectal cancer model [53].

Overall, our study demonstrated that specific doses of resveratrol exhibit hormetic antitumour activity in a cell-line-dependent manner. Moreover, we identified the cellular targets of resveratrol in OVCAR-8 and SKOV-3 cells, which include the reduction in p-NF-κB and SLUG protein levels at a single-cell level, resulting in a significant decrease in cancer cell invasion. However, it is important to note that this study has limitations, such as the utilisation of a small panel of ovarian cancer cell lines, and the bioavailability of resveratrol in chicken embryo CAMs is unknown. Therefore, incorporating additional ovarian cancer cell lines and analysing resveratrol concentrations in CAMs would enhance the justification for considering resveratrol as an alternative self-managed complimentary treatment.

## 4. Materials and Methods

### 4.1. Cell Culture

Two ovarian cancer cell lines, OVCAR-8 and SKOV-3, were obtained from Dr. Judith McKenzie of the Haematology Research Group at the University of Otago, Christchurch, New Zealand. These cell lines were cultured in DMEM media (Life Technologies, Auckland, New Zealand), supplemented with 5% fetal bovine serum (FBS) (Life Technologies, New Zealand) and 5% Serum Replacement Media (Life Technologies, New Zealand), along with Pen/Strep (Life Technologies, New Zealand) at a working concentration of 100 units/mL penicillin, 100 units/mL streptomycin, 2 mM Gluta-MAX (Life Technologies, New Zealand), and 1 µg/mL Fungizone (Life Technologies, New Zealand). The final glucose concentration in the media was 5.5 mM. This supplemented media is referred to as working media hereafter. Cells were maintained at 37 °C in a humidified 5% CO_2_ atmosphere. Authentication of OVCAR-8 and SKOV-3 cell lines was conducted using short tandem repeat (STR) testing by CellBank (Children’s Medical Research Institute, Sydney, NSW, Australia).

### 4.2. Three-Dimensional (3D), Chicken Embryo Model and Resveratrol Treatment

OVCAR-8 and SKOV-3 cells at a concentration of 200,000 cells were combined with a 60 µL mixture of collagen I (2 mg/mL) and GelTrex (25% vol/vol). The cell mixtures were then transferred to sterile parafilm-coated plates and allowed to polymerise in the cell incubator for 25 min. Once polymerised, the gels were cultured in a 24-well plate with cell culture media for 4 days, with fresh media replaced every two days. These cells embedded in collagen gels were subsequently used for implantation on the CAM.

Chicken eggs, comprising a mix of Orpington and Dorking chicken breeds, were procured from a local farm. The eggs underwent two washes with warm water followed by treatment with a diluted disinfection solution (Brinsea, Weston-super-Mare, UK). Subsequently, the eggs were submerged in a 2 mM CuSO_4_ solution for 5 min and left to dry. Once dried, the eggs were cleaned with an iodine solution and left to dry overnight. The eggs were then transferred to an incubator set at 37 °C with 60% humidity. From day 1 to day 4, the eggs were rotated twice daily. On the 4th day, a small square window measuring 2 cm × 2 cm was created using a drill on the air pocket side of the eggs. The eggshells were removed, and sterile plastic sticky tape was used to seal the window. The eggs were cleaned again with the iodine solution and returned to the incubator until the 7th day. On the 7th day, two collagen gels containing cancer cells were placed on top of the CAM, and the window was resealed before placing the eggs back in the incubator until the 12th day. On the 12th day, designated doses of resveratrol totalling 200 µL were applied to the CAM, away from the tumour Implants. Fresh resveratrol was replaced every two days until the 18th day. A stock solution of resveratrol (2 mg/mL) was prepared using a 5.5 mM glucose solution. Working doses of resveratrol were calculated and mixed with the glucose solution to achieve a final volume of 200 µL. The control group was supplied with a 5.5 mM glucose solution.

### 4.3. Collections of Tumour CAMs and Histological and Immunohistological Analysis

At day 18, tumour implants were excised using a clean pair of scissors, either placed in lysis buffer or fixed in 4% paraformaldehyde, and then kept at 4 °C for sectioning. The fixed tumour implants underwent washing with 1xPBS before being embedded vertically in optimal cutting temperature (OCT) compound (ThermoScientific, Auckland, New Zealand), ensuring the absence of air bubbles. The samples were frozen at −80 °C overnight before being transferred to −20 °C prior to sectioning with a Cryostat (Leica CM1860, Leica Biosystems, Nußloch, Germany). Sections of the tumour implants, 7 µm in thickness, were captured on Super-Frost Plus slides and stored at −20 °C for histological and immunohistochemical staining. Some slides underwent staining with hematoxylin-eosin (H and E), while others were stained with antibodies at a 1/200 dilution for VEGF (catalogue number SC152, Santa Cruz Biotechnology, Dallas, TX, USA), NF-κB (catalogue number SC372, Santa Cruz Biotechnology, USA), pNF-κB (catalogue number SC33020, Santa Cruz Biotechnology, USA), and SLUG (catalogue number PRS3959, Merck, Auckland, New Zealand). Immunostaining of VEGF, NF-κB, and pNF-κB was detected using the Mouse and Rabbit Specific HRP/AEC (ABC) Detection IHC Kit (catalogue number Ab93705, Abcam, Auckland, New Zealand). Immunostaining of SLUG protein was detected using the fluorescent method; slides were stained with a 1/1000 dilution of Slug antibody and a 1/500 dilution of anti-Rabbit FITC (Merck, Auckland, New Zealand). Then, the slides were mounted with an anti-fading solution (2 mg/mL phenylene diamine prepared in 50% (vol/vol) glycerol in 1xPBS pH 8)) before imaging with a fluorescent microscope. Tumour sections were imaged using a Zeiss Axioimager Z1 microscope (AxioVision 4.5, Carl Zeiss Oberkochen, Germany).

### 4.4. Protein Analysis with Western Blot

The tumour CAMs were rinsed twice with a cold 1xPBS solution to remove red blood cells from the embryos. Subsequently, they were treated with 0.5 mL of lysis buffer containing 0.1% SDS, 20% glycerol, 1xPBS, a protease inhibitor, a phosphatase inhibitor, and 1 mM Na_3_VO_4_. Following this, 50 µL of sample buffer composed of 0.2% (vol/vol) bromophenol blue, 25% (vol/vol) glycerol, and 10% SDS in Tris-HCl at pH 6.8 was added, and the protein lysates were boiled for 10 min. Prior to loading, the cell lysates were thoroughly mixed and centrifuged at 9700× *g* for 5 min. Subsequently, 10–30 μg of protein lysate was loaded and separated by SDS-PAGE using a 7% stacking gel and a 10% separating gel, running at 120 volts with a Tris-glycine running buffer. Precision Plus Dual Color and Precision Plus Protein Western C SDS-PAGE markers were employed. The proteins were then electrophoretically transferred onto polyvinylidene difluoride (PVDF) membranes in ice-cold transfer buffer (0.006% SDS, 25 mmol/L Tris base, 200 mmol/L glycerin) at 100 V for 60 min. The PVDF membranes were treated with either 5% skim milk (PAMS, Auckland, New Zealand) or 4% BSA (Merck, Auckland, New Zealand) for 60 min, followed by overnight incubation at 4 °C with relevant primary antibodies. After extensive washing with 20 mM Tris-HCl pH 7.4 plus 0.1% Tween-20, the membranes were incubated with anti-Rabbit-HRP (1/5000) or anti-Mouse-HRP (1/5000) (Merck, Auckland, New Zealand) mixed with 1/10,000 DF of Precision Protein Strep Tactin-HRP Conjugate (Bio-Rad, Auckland, New Zealand) for 90 min. The membranes were developed using Clarity™ Western ECL substrate (Bio-Rad, New Zealand), and protein bands were visualised and subjected to densitometry analysis using Alliance 4.7, Unitec (Cambridge, UK). All original Western blot images can be found in Appendix A.

### 4.5. Imaging Analysis

Tumour CAMs were dissected from the embryos and positioned upside-down to visualise blood vessels in a PBS solution using a stereo microscope. Histological staining was performed to visualise red blood cells in tumour CAMs and invasive tumour cells within CAM areas, with subsequent quantification of red blood cells and tumour cells using Adobe Photoshop 2021. Immunohistological staining of VEGF, NF-κB, and p-NF-κB was analysed using ImageJ FIJI 1.53m (Wayne Rasband and contributors National Institutes of Health, Bethesda, MD, USA). Additionally, immunofluorescence staining of the SLUG protein in tumour CAMs was quantified using ZEN 3.3 (Blue Edition, Carl Zeiss processing software). A two-way analysis of variance (ANOVA) and student’s *t*-tests from GraphPad Prism 10 software was applied to analyse the data.

## Figures and Tables

**Figure 1 ijms-25-04374-f001:**
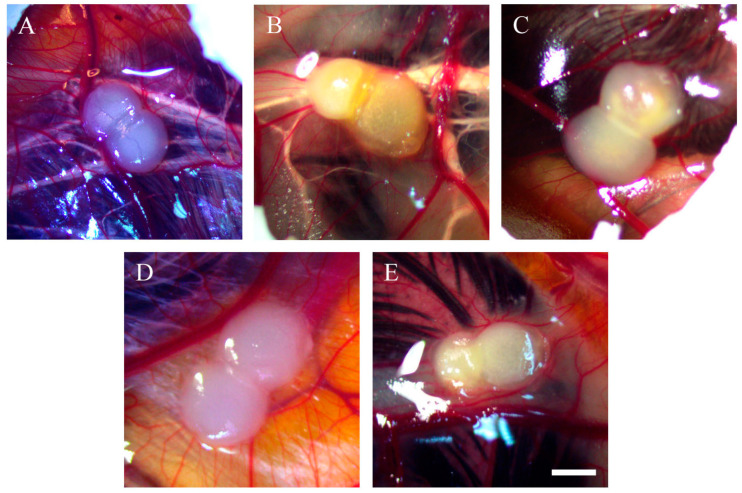
Stereomicroscopic images of OVCAR-8 tumour implants after 18 days of growth on CAMs. Tumour implants were treated with various doses of resveratrol for six days, starting on day 12 of embryonic development, and the images were captured. Control (**A**), 0.228 µg (5 µM) (**B**), 0.456 µg (10 µM) (**C**), 45.6 µg (1 mM) (**D**), 91.24 µg (2 mM) (**E**). Scale bar, 1 mm.

**Figure 2 ijms-25-04374-f002:**
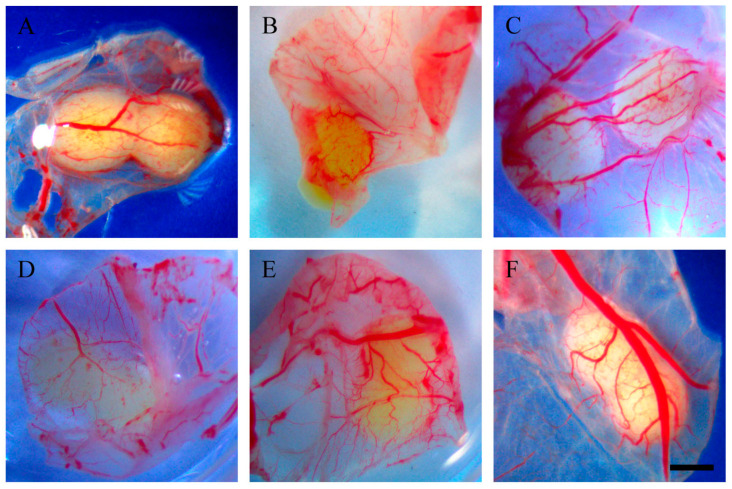
Stereomicroscopic images of excised and inverted OVCAR-8 tumour implants after 18 days of growth on CAMs. Tumour implants were excised from the surrounding CAM and inverted so the vascular networks could be viewed from underneath. Control (**A**), 0.228 µg (5 µM) (**B**), 0.456 µg (10 µM) (**C**), 45.6 µg (1 mM) (**D**), 91.24 µg (2 mM) (**E**), 182.4 µg (4 mM) resveratrol (**F**). Scale bar, 1 mm.

**Figure 3 ijms-25-04374-f003:**
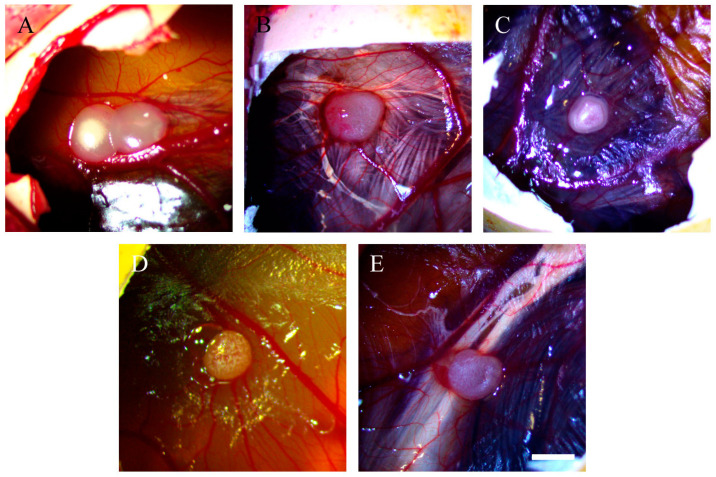
Stereomicroscopic images of SKOV-3 tumour implants after 18 days of growth on CAMs. Tumour implants were with various doses of resveratrol for six days, starting at day 12 of embryonic development, and then images were captured. Control (**A**), 0.228 µg (5 µM) (**B**), 0.456 µg (10 µM) (**C**), 45.6 µg (1 mM) (**D**), and 91.24 µg (2 mM) resveratrol (**E**). Scale bar, 1 mm.

**Figure 4 ijms-25-04374-f004:**
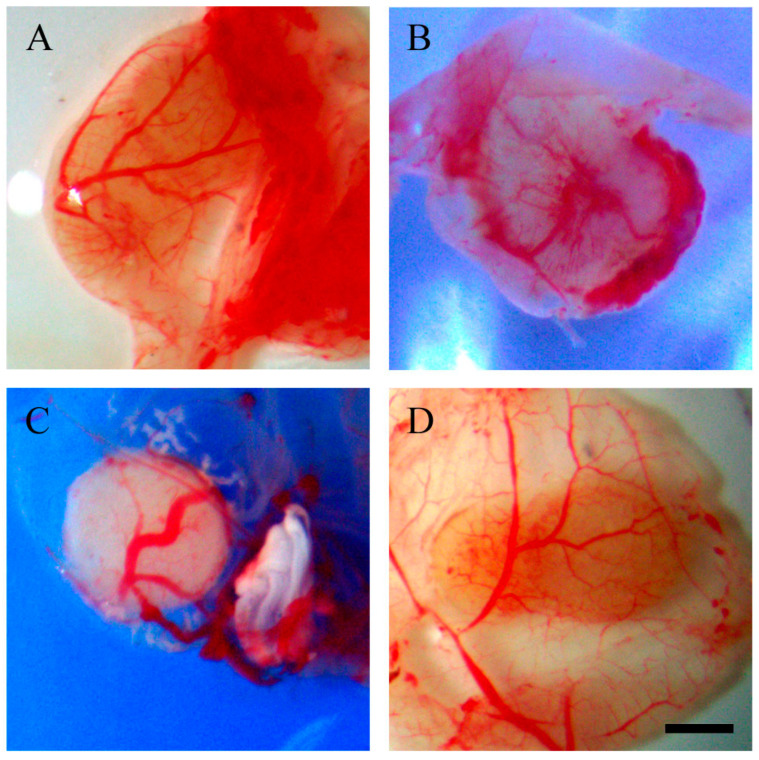
Stereomicroscopic images of excised and inverted SKOV-3 tumour implants after 18 days of growth on CAMS. Tumour implants were excised from the surrounding CAM and inverted so the vascular networks could be viewed from underneath. Control (**A**), 0.228 µg (5 µM) (**B**), 45.6 µg (1 mM) (**C**), 91.24 µg (2 mM) resveratrol (**D**). Scale bar, 1 mm.

**Figure 5 ijms-25-04374-f005:**
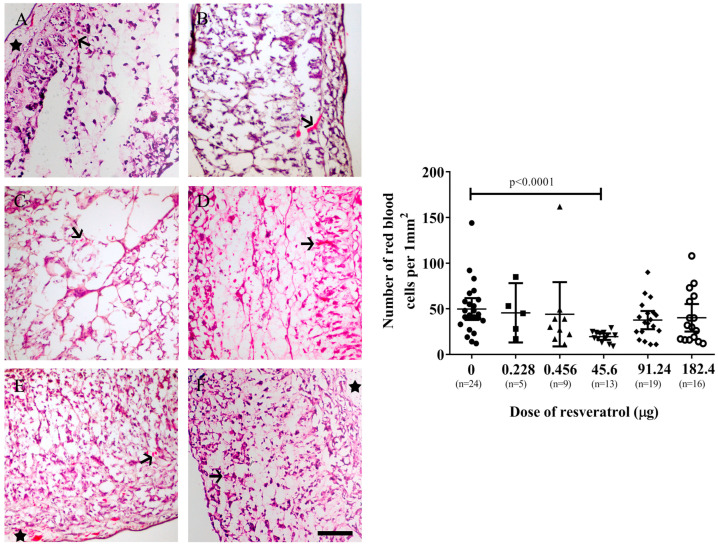
**Left**: Microscopic images of OVCAR-8 tumour implants after six days of treatment with various doses of resveratrol ((**A**) control, (**B**) 0.228 µg (5 µM), (**C**) 0.456 µg (10 µM), (**D**) 45.6 µg (1 mM), (**E**) 91.24 µg (2 mM), (**F**) 182.4 µg (4 mM)). Red blood cells are in red (arrows), and CAM areas are indicated by stars. **Right**: The number of red blood cells was counted and plotted using various doses of resveratrol. Data are expressed as means ± SEM, *n* = number of sectioned tumour implants, which were from at least five embryos. Data considered statistically significant compared to controls are indicated as *p* < 0.05, *t*-student test. The scale bar is 200 µm.

**Figure 6 ijms-25-04374-f006:**
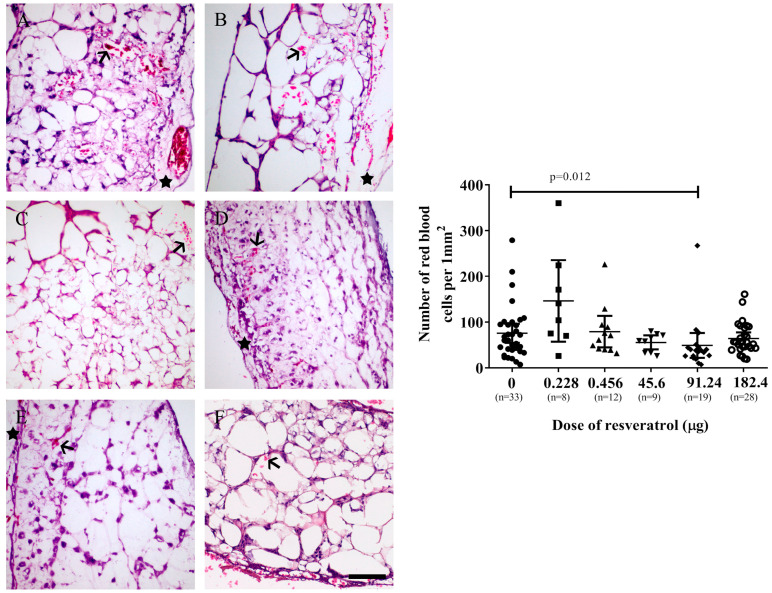
**Left**: Microscopic images of SKOV-3 tumour implants after six days of treatment with various doses of resveratrol ((**A**) control, (**B**) 0.228 µg (5 µM), (**C**) 0.456 µg (10 µM), (**D**) 45.6 µg (1 mM), (**E**) 91.24 µg (2 mM), (**F**) 182.4 µg (4 mM)). Red blood cells are in red (arrows), and CAM areas are indicated by stars. **Right**: The number of red blood cells that were counted and plotted with various doses of resveratrol. Data are expressed as means ± SEM, *n* = number of sectioned tumour implants, which were from at least five embryos. Data considered statistically significant compared to controls are indicated as *p* < 0.05, *t*-student test. The scale bar is 200 µm.

**Figure 7 ijms-25-04374-f007:**
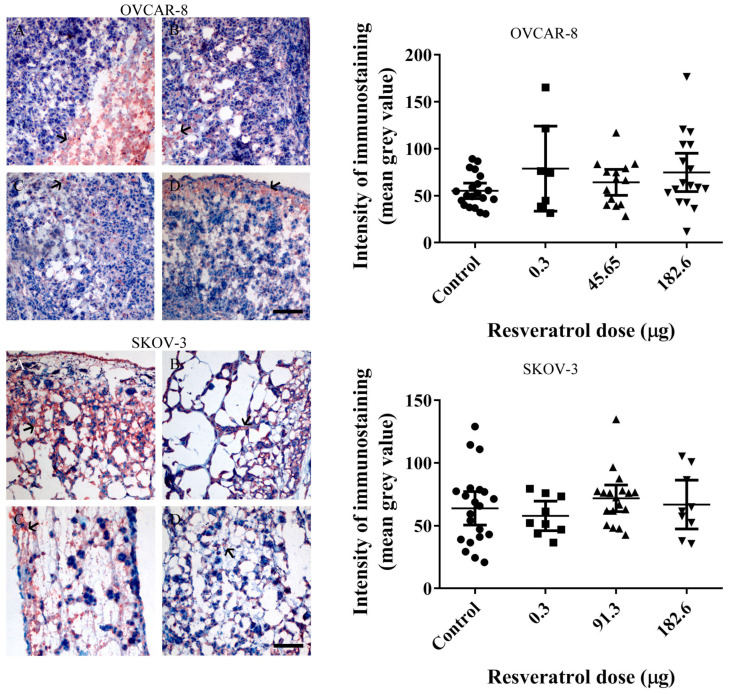
Immunohistological staining of vascular endothelial growth factor (VEGF) in tumour sections. Left-microscopic images of OVCAR-8 (**Top:** (**A**) control, (**B**) 0.3 µg (5 µM), (**C**) 91.3 µg (2 mM), (**D**) 182.6 µg (4 mM)) and SKOV-3 (**Bottom:** (**A**) control, (**B**) 0.3 µg (5 µM), (**C**) 91.3 µg (2 mM), (**D**) 182.6 µg (4 mM)) tumour implants were immune-stained with an anti-VEGF antibody. The red-brown colour represents levels of VEGF protein (arrows). The intensity of immunostaining of anti-VEGF antibody was subjected to analysis using Fiji ImageJ software 1.53m, and the densitometry of grey value was calculated and plotted. Data are expressed as means ± SEM. The scale bar is 200 µm.

**Figure 8 ijms-25-04374-f008:**
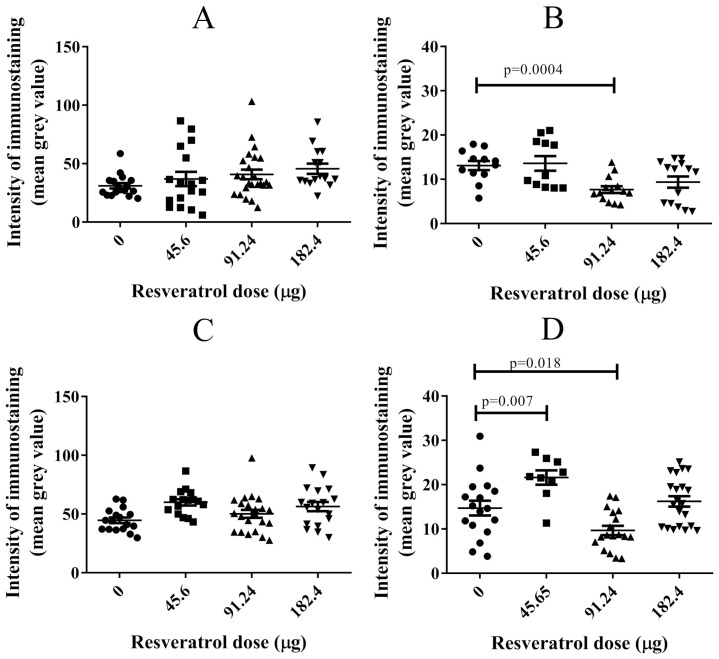
Immunohistological staining of NF-κB (**A**,**C**) and pNF-κB (**B**,**D**) in OVCAR-8 (**A**,**B**) and SKOV-3 (**C**,**D**) tumour implants treated with three doses of resveratrol (45.6 (1 mM), 91.24 (2 mM) and 182.4 µg (4 mM)) for six days. Tissue implants were sectioned and stained with anti-NF-κB and phospho-NF-κB antibodies. The histological immunostaining of the antigens was quantitative using Fiji ImageJ software 1.53m. Data are expressed as means ± SEM. Data considered statistically significant compared to controls are indicated as *p* < 0.05, *t*-student test.

**Figure 9 ijms-25-04374-f009:**
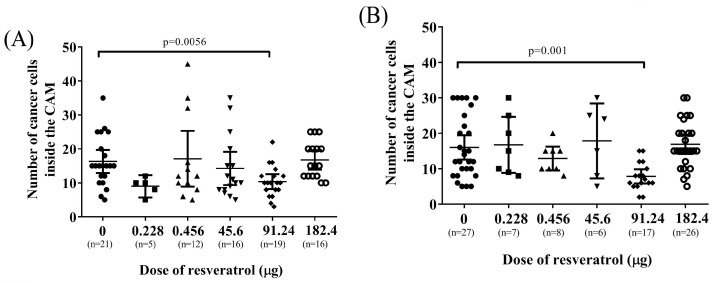
The number of cancer cells in the area of mesoderm layers of CAM implanted with OVCAR-8 cell line (**A**) and SKOV-3 (**B**) treated with various doses of resveratrol. Tumour implants were treated with different doses of resveratrol, 0.228 µg (5 µM), 0.456 µg (10 µM), 45.6 µg (1 mM), 91.24 µg (2 mM), and 182.4 µg (4 mM). Data are expressed as means ± SEM, *n* = number of sectioned tumour implants, which were from at least five embryos. Data considered statistically significant compared to controls are indicated as *p* < 0.05, *t*-student test.

**Figure 10 ijms-25-04374-f010:**
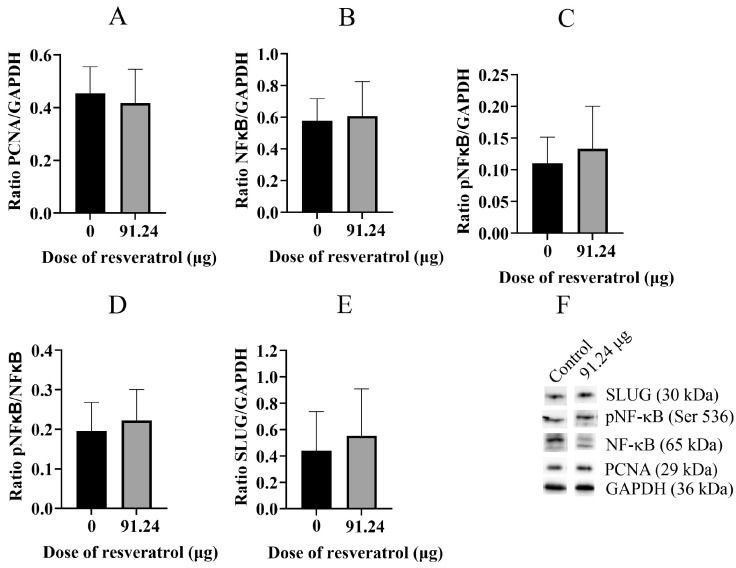
Western blot analysis of specific proteins extracted from whole tumour CAM tissue of the OVCAR-8 cell line. The ratios of protein band intensities for PCNA/GAPDH (**A**), NFκB/GAPDH (**B**), pNFκB/GAPDH (**C**), pNFκB/NFκB (**D**), and SLUG/GAPDH (**E**) are presented. Bands of each protein were analysed by Western Blotting (**F**). A resveratrol dose of 91.24 µg (2 mM) was selected for the inhibition study with tumour implants.

**Figure 11 ijms-25-04374-f011:**
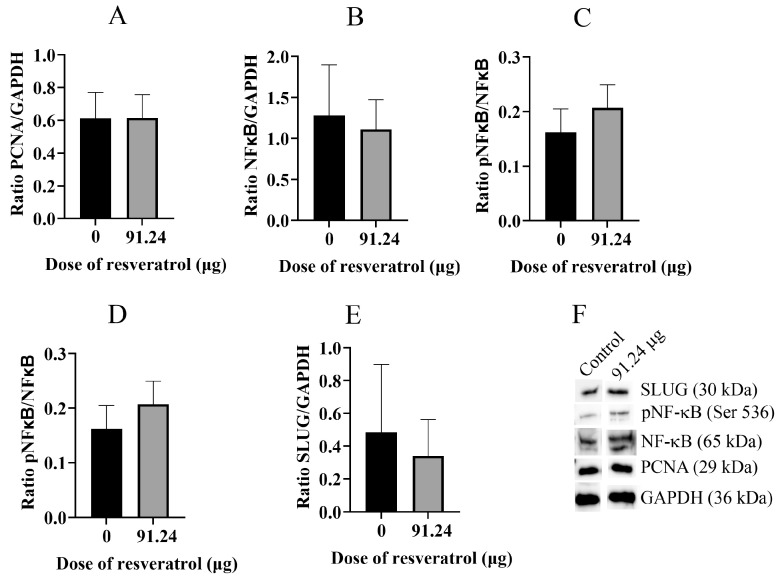
Western blot analysis of specific proteins extracted from whole tumour CAM tissue of the SKOV-3 cell line. The ratios of protein band intensities for PCNA/GAPDH (**A**), NFκB/GAPDH (**B**), pNFκB/GAPDH (**C**), pNFκB/NFκB (**D**), and SLUG/GAPDH (**E**) are presented. Bands of each proteins were analysed by Western Blotting (**F**). A resveratrol dose of 91.24 µg (2 mM) was selected for the inhibition study with tumour implants.

**Figure 12 ijms-25-04374-f012:**
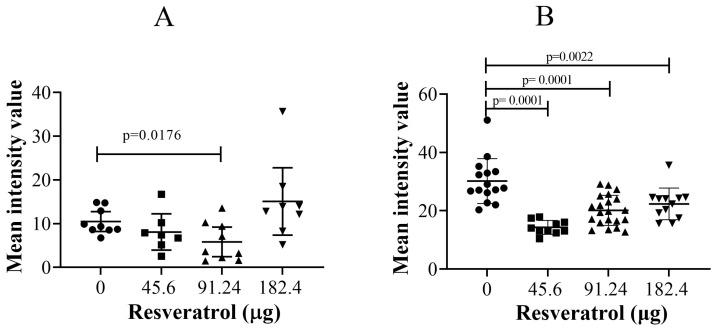
Immunohistochemical staining of SLUG antigens in OVCAR-8 (**A**) and SKOV-3 (**B**) tumour implants treated with three doses of resveratrol (45.6 (1 mM), 91.24 (2 mM), and 182.4 (4 mM) µg) for six days. The immunostaining of the antigens was quantitated using Fiji ImageJ software 1.53m. Data considered statistically significant compared to controls are indicated as *p* < 0.05, *t*-student test.

## Data Availability

The datasets are available from the corresponding author upon request.

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
