# Peer review of "Effects of Resveratrol on In Vivo Ovarian Cancer Cells Implanted on the Chorioallantoic Membrane (CAM) of a Chicken Embryo Model"

_ijms, 2024, doi:10.3390/ijms25084374_

Round 1

Reviewer 1 Report

Comments and Suggestions for Authors

The manuscript submitted by Chitcholtan et al. presents a very interesting study aiming to dissect the in vivo antitumor effects of the nutraceutical resveratrol in a innovative model such as implants on the Chorioallantoic Membrane (CAM) of a chicken embryo. The topic is appealing and overall the results of the study are convincing.

However, the authors should address some concerns:

1) the hormetic nature of resveratrol should be discussed more in detail in the discussion. Authors cited some literature but this aspect should be extended with relevant recent citations. The hormesis may explain also the controversial effects observed with different concentrations of resveratrol in this study.

2) the beneficial antitumor effects of resveratrol in vivo are limited by its poor availability. Authors should discuss this aspect and speculate whether this model may experience the same limitations in bioavailability reported in other preclinical models. 

3) Authors may convert the dosage of resveratrol used in their experimental plan in concentrations (micromolarity and so). This may help the discussion and comparisons of the effects observed in their study and the findings in the literature. 

4) The presentation of the western blotting in Figure 10F and Figure 11F is not methodologically correct. Authors should provide the full western blotting without cuttings in with the lanes are not discontinued.

5) I have a concern in the interpretation of Figure 10 and 11. Authors showed that resveratrol reduced p-NFkB levels in immunostaining (Figure 8). How do they explain the lack of effects detected by western blotting?

6) In a similar way, how the authors justify the lack of modulation on PCNA levels? They may assess other proliferation markers (e.g. Ki-67) to confirm their results.

7) line 195: "Figures 10 and 11 illustrate that the total protein levels of SLUG were not reduced, which may be attributed to the presence of SLUG in the CAM itself.". Is there any possibility to distinguish the proteins coming from CAM and the ones from the implant? Collect the cell homogenates only from the implant?

8) Authors should disclose more in depth the state-of-art of the preclinical evidence of benefical effects of resveratrol, citing relevant and update literature.

Comments on the Quality of English Language

The quality of English language is fine, only few minor editing are required to improved the readability of the manuscript.

Reviewer 2 Report

Comments and Suggestions for Authors

This manuscript submitted by Katia Kenny Chitcholtan et al. presents “Effects of Resveratrol on in vivo Ovarian Cancer Cells Implanted on the Chorioallantoic Membrane (CAM) of a Chicken Embryo Model” 

The manuscript presents lots of research data but too much data even makes a hard to follow the story. And story's conclusion was not clear, so it was difficult to catch up with the author's point. The current manuscript needs to improve the data presentation and the sentence for the general reader.

It is not sure whether Resveratrol affects the CAM system. The evidence is not enough to clear conclusion. Many experiments showed different amounts of Resveratrol were applied to the CAM system, but it did not consistently inhibit cell growth, or reduce angiogenesis even did not inhibit the inflammation pathway too.

Also, not clear how many eggs are used for each condition (dose-dependent manner of Resveratrol).

For the vascular networks, the current protocol designed for Resveratrol treatment was 12 days of CAM, Figures 1-4 show that no effect for inhibit of vascular networks at all.

So why not start at day 6 and check to block the angiogenesis or not by Resveratrol dose?

How about giving to a final concentration such as mM or micro Mol instead amount which is present on bar graph?

Line 85: “tumours in ovarian cancer patients (Figure 2).” Figure 2 is not a patient’s data so this sentence should remove.

Figures 5 and 6: Lable was too small to identify and nor clear, please increase the size. What is the size of the scale bar?  

What is it n=24, n=5 n=9, n=13 n=19, and n=16 below drug amount on the bar graph?

Wondering, is just a different field of image provided from a single experiment.

Only one dose (each 45.6 ug and 91.24 ug) of the drug inhibits blood cell numbers, logically it is hard to understand even if this is not the highest dose of a drug. So, authors may need more evidence for those conclusions. So hard to tell that “VEGF may not play a major role in the angiogenesis of tumour implants in the CAM model”

Line 145, 146: the data is not significantly different so should remove this sentence “ Data considered statistically significantly compared to controls are  indicated as p< 0.05, t-student test.

Figures 10 and 11 will be a key data for this manuscript but both data present looks like Resveratrol has no affect al all for NF-kB, phospho NF-kB, and Slug expression. But authors clam that Resveratrol inhibit SLUG expression.  The data is not consistent with Figure 12. So, It is not sure Resveratrol regulate SLUG or not.

 In general, western blot data with pick and crop the band from the blot is not a good way to presentation of data. 

Round 2

Reviewer 1 Report

Comments and Suggestions for Authors

The revision of the manuscript satisfies the issues.